# Feasibility and Radiological Outcome of Minimally Invasive Locked Plating of Proximal Humeral Fractures in Geriatric Patients

**DOI:** 10.3390/jcm11226751

**Published:** 2022-11-15

**Authors:** Konrad Schuetze, Alexander Boehringer, Raffael Cintean, Florian Gebhard, Carlos Pankratz, Peter Hinnerk Richter, Michael Schneider, Alexander M. Eickhoff

**Affiliations:** Center of Surgery, Department of Traumatology, Hand-, Plastic-, and Reconstructive Surgery, University of Ulm, 89075 Ulm, Germany

**Keywords:** proximal humerus fractures, geriatric patients, cement augmentation, modern implant for proximal humerus, complications in proximal humerus fractures

## Abstract

Background: Proximal humerus fractures are common injuries in the elderly. Locked plating showed high complication and reoperation rates at first. However, with second-generation implants and augmentation, minimally invasive locked plating might be a viable alternative to arthroplasty or conservative treatment. Material and Methods: A retrospective chart review was performed for all patients with proximal humerus fractures treated between 2014 and 2020 with locked plating. All patients over 60 years of age who underwent surgery for a proximal humerus fracture with plate osteosynthesis (NCB, Philos, or Philos with cement) during the specified period were included. Pathological fractures, intramedullary nailing, or arthroplasty were excluded. Primary outcome measurements included secondary displacement and surgical complications. Secondary outcomes comprised function and mortality within one year. Results: A total of 249 patients (mean age 75.6 +/− 8.9 years; 194 women and 55 men) were included in the study. No significant difference in the AO fracture classification could be found. Ninety-two patients were surgically treated with first-generation locked plating (NCB, Zimmer Biomet, Wayne Township, IN, USA), 113 patients with second-generation locked plating (Philos, Depuy Synthes, Wayne Township, IN, USA), and 44 patients with cement-augmented second-generation locked plating (Philos, Traumacem V+, Depuy Synthes). A 6-week radiological follow-up was completed for 189 patients. In all groups, X-rays were performed one day after surgery, and these showed no differences concerning the head shaft angle between the groups. The mean secondary varus dislocation (decrease of the head shaft angle) after six weeks for first-generation locked plating was 6.6 ± 12° (*n* = 72), for second-generation locked plating 4.4 ± 6.5 (*n* = 83), and for second-generation with augmentation 1.9 ± 3.7 (*n* = 35) with a significant difference between the groups (*p* = 0.012). Logistic regression showed a significant dependency for secondary dislocation for the type of treatment (*p* = 0.038), age (*p* = 0.01), and preoperative varus fracture displacement (*p* = 0.033). Significantly fewer surgical complications have been observed in the augmented second-generation locked plating group (NCB: 26.3%; Philos 21.5%; Philos-augmented 8.6%; *p* = 0.015). Range of motion was documented in 122 out of 209 patients after 3 months. In the Philos-augmented group, 50% of the patients achieved at least 90° anteversion and abduction, which was only about a third of the patients in the other 2 groups (NCB 34.8%, *n* = 46; Philos 35.8%, *n* = 56; augmented-Philos 50.0%, *n* = 20; *p* = 0.429). Conclusion: Minimally invasive locked plating is still a valuable treatment option for geriatric patients. With augmentation and modern implants, the complication rate is low and comparable to those of reverse shoulder arthroplasty reported in the literature, even in the challenging group of elderly patients.

## 1. Introduction

Fractures of the proximal humerus are common injuries, especially in women above the age of 60. In an aging population, an even higher incidence of these fracture can be expected. Rupp et al. reported an increase of proximal humerus fractures of 10% between 2009 and 2019 in Germany. Reasons for this include a higher incidence of osteoporosis and the increasing number of falls [1,2]. Determining the best treatment for proximal humerus fractures remains a challenge. In the case of undisplaced or minimally displaced fractures, a non-operative procedure is favorable and the most common [3]. Even in displaced fractures, no significant difference in regards to function and quality of life could be found between an operative and non-operative treatment [4]. However, surgery is recommended for displaced and three- or four-part fractures. Depending on the fracture morphology, locked nailing, locked plating, or even shoulder arthroplasty with a reverse prosthesis are common options [5,6]. Locking nail systems do not appear to be an option for complex fractures involving a displacement of the tuberosities [6]. Primary reverse shoulder arthroplasty (RSA) is often chosen when a significant reduction and stable fixation are not achievable and the vascularity of the humeral head is damaged [7].

Compared to RSA, surgical complications in locking plates, such as the loss of reduction and cut outs, are common, with revision rates up to 32% [8,9]. However, similar functional results are shown one year after surgery [9], and high revision rates were observed in first-generation implants. Due to the high complication rate, various surgical options have been developed, such as inferomedial screws or fibula allograft augmentation [10]. Many studies are investigating complications and clinical results after conservative treatment, plating of proximal humerus fractures, and primary reverse shoulder arthroplasty, but only one clinical study is available evaluating complications and clinical results after screw tip augmentation [11]. The first hypothesis of this investigation argues that screw tip cement augmentation reduces the risk of the implant loosening in geriatric proximal humerus fractures. After cement augmentation, no influence on the one-year mortality figures is expected (second hypothesis).

## 2. Methods

The study was approved by the institutional ethical committee under the number, 169/20-FSt/TR. The study was a retrospective cohort study at a level one trauma center. All patients over the age of 60 years with a proximal humerus fracture treated between January 2014 and December 2020 were included in the study (Table 1).

Indications for surgery included the translation of the humerus shaft and a multifragmentary fracture morphology with displacement.

The type of osteosynthesis was chosen individually by availability, the personal preference of the surgeon (NCB vs. Philos), and subjective quality of the bone (Philos vs. Philos with screw tip augmentation).

Ninety-two patients were treated with first-generation locked plating (NCB Proximal humerus plating, Zimmer Biomet, Wayne Township, IN, USA) from 2014 to 2018, 113 with second-generation locked plating (Philos, Depuy Synthes, Wayne Township, IN, USA, 2014–2020) and 44 with augmented second-generation locked plating (Philos + Traumacem V+, Depuy Synthes, 2018–2020). Mean time to surgery was 3.9 +/− 4.4 days.

Surgery was performed in beach chair position and traction was applied with a Trimano Fortis (Arthrex, Port of Naples, FL, USA) fixed to the arm (Figure 1).

After a minimally invasive delta split procedure, traction sutures were applied through the rotator cuff. A reduction was achieved by the traction and manipulation of the sutures. In some cases, a reduction was achieved by the direct manipulation of the humeral head through the fracture. After a provisional fixation was achieved with 1.6 mm K-wires, the plate was placed and fixed with four to seven screws in the head and three screws in the diaphysis (Figure 2).

After fluoroscopic control of reduction and screw placement, a radiological contrast agent was injected prior to augmentation to prevent leakage into the joint. Afterwards, every screw was augmented with 0.5–1 mL of Traumacem V+ under fluoroscopic control (Figure 3).

The arm was immobilized for 10 days after surgery. Physiotherapy with passive forward flexion and abduction up to 90° started on day 11. Lifting and free motion was allowed starting week 7 after surgery.

The head-shaft angle (HSA) was measured by a resident and a consultant radiologically preoperatively, postoperatively, and after 6 weeks in all fractures except A1 fractures (Figure 4). Primary outcome measures included secondary dislocation and surgical complications. Secondary outcome measures included function after 3 months, mortality, and discharge disposition. A one-year follow up to obtain information about the mortality and complications was performed by analyzing the electronic record of the patient. In the geriatric and partly immobilized collective, many patients were not seen as outpatients. In these cases, follow up was performed by phoning the patients.

Data analysis was performed with IBM SPSS Statistics (V21.0, SPSS Inc. Chicago, IL, USA) and Microsoft Excel (V16.3, Microsoft Cooperation, Redmond, WA, USA). Demographic characteristics are described as mean and standard deviation. Group comparisons via the chi-square test were used to compare secondary dislocations between the groups. For further analysis, a logistic regression was performed for all variables related to the secondary dislocation and surgical complications. Group comparisons via Chi-square test were performed for number of revision surgeries and rate of secondary arthroplasty.

## 3. Results

### 3.1. Patient Population

For 249 patients, medical records were reviewed. Out of these 249 patients, 55 were male and 194 were female. The mean age was 75.6 +/− 8.9 years. Eight patients were classified as ASA I, 66 as ASA II, 154 as ASA III, and 21 as ASA IV. The mean time to surgery was 3.9 +/− 4.4 days, and the mean surgery time was 69.5 +/− 66.1 min. After the hospital discharge, 40 patients were lost to follow up, and the remaining 209 patients were followed up to one year. Twenty out of these 209 patients had a secondary dislocation within the first 6 weeks and were, therefore, excluded from the 6-week radiographical follow up. Overall, out of the 249 patients, radiological follow up was possible for 189 patients after 6 weeks.

### 3.2. Fracture Classification

All fractures were classified according to the AO-classification. Five fractures were classified as 11-A1, 50 as A2, 10 as A3, and 89 as B1. C1 fractures occurred in 17 and C3 fractures in 88 patients. Ninety-three patients showed a varus displacement with an HSA smaller than 135°, while 149 patients showed a valgus displacement with an HSA > 135°. A1 fractures showed no varus/valgus dislocation. Fracture classification and type of treatment for all patients is shown in Figure 5.

### 3.3. Radiological Outcome

The mean postoperative HSA was 132.4 +/− 9.7°. One hundred and fifty-five cases had a mean postoperative varus HSA of 8.5 +/− 7.2°. Ninety-nine cases had a mean postoperative valgus HSA of 6.4 +/− 5.1°. There was no difference between the postoperative HSA between the 3 groups, showing a comparable quality of reduction. After 6 weeks, 189 patients had a radiological follow up. The mean secondary dislocation for first-generation locked plating was 6.6 +/− 12° (*n* = 72, 38%), for second-generation locked plating, 4.4 +/− 6.5° (*n* = 83, 43.9%), and for second-generation with augmentation, 1.9 +/− 3.7° (*n* = 35, 18.5%). The difference between the groups was significant (*p* = 0.012). Logistic regression showed a significant dependency for the secondary displacement for the type of treatment (*p* = 0.038), age (*p* = 0.01), and preoperative varus fracture displacement (*p* = 0.033). Varus displaced fractures had a 6-fold increased risk for secondary displacement. For every year above 60 years of age, the risk for secondary displacement increased by a factor of 1.1. The mean secondary displacement after 6 weeks is shown in Figure 6.

### 3.4. Complications

Surgical complications occurred in 44 (21%) out of 209 patients that could be followed up to one year. Varus collapse, defined as a varus displacement leading to surgery, occurred in 31 patients; 5 patients had other secondary displacements; 2 patients had implant related infections; and 6 patients had primary screw misplacements. Surgical complications occurred significantly less in the second-generation locked plating with augmentation group (NCB: 26.3%; Philos 21.5%; Philos-augmented 8.6%; *p* = 0.015). Logistic regression showed no dependency for age, AO-classification, or pre- or postoperative HSA. Severe cases of varus collapse were observed only in the NCB and Philos groups without cement augmentation. Twelve patients needed revision surgery with implant removal in 4 cases (NCB 2.5%; Philos 2.1%; *p* = 0.102) and reverse shoulder arthroplasty in 8 cases (NCB 6.3%; Philos 3.2%; *p* = 0.345), with no significant difference between the groups.

### 3.5. Discharge Disposition and Mortality

The mean hospital stay was 7.4 +/− 3.2 days. Overall, 86.7% (*n* = 216) of the patients could be discharged home or to inpatient rehabilitation. Thirty patients were discharged to a nursing home, and 3 patients died during the hospital stay. Eighteen out of 209 patients (8.6%) that were followed up for 1 year died within the first year after surgery.

### 3.6. Function

The range of motion was only documented in 122 out of 209 patients after 3 months. In the Philos-augmented group, 50% of the patients achieved at least 90° anteversion and abduction, which was only about a third of the patients in the other 2 groups (NCB 34.8%, *n* = 46; Philos 35.8%, *n* = 56; augmented-Philos 50.0%, *n* = 20; *p* = 0.429). There was no significant difference between the groups.

## 4. Discussion

Proximal humerus fractures are commonly treated conservatively. However, in complex fracture situations, surgery is necessary. Knowing of the adverse events after the plating of proximal humerus fractures, especially in geriatric patients, primary reverse arthroplasty is becoming more of a common treatment option, accompanied with a more invasive surgery [3]. The aim of this study was to prove the feasibility of minimally invasive, modern locked plating for the treatment of proximal humerus fractures in geriatric patients. For this purpose, two generations of locking plates with or without cement augmentation were compared. In this study, second-generation locked plating with cement augmentation was found to be superior to first-generation locked plating and second-generation without augmentation regarding the secondary displacements. An overall surgical complication rate of 21% was found, occurring significantly less frequen when the second-generation locked plate with augmentation was used (NCB: 26.3%; Philos 21.5%; Philos-augmented 8.6%; *p* < 0.05).

In surgery for proximal humerus fractures, an especially feared complication is the varus collapse, which occurred in 31 out of 209 patients in this study but only in the NCB and Philos groups.

The biomechanical studies of Unger et al., Röderer et al., and Scola et al. support the advantages of augmentation, particularly, a delayed varus collapse and significant more load cycles until failure [12,13,14]. There are only a few clinical studies for screw augmentation. Katthagen et al. investigated a smaller cohort of 24 patients after screw tip augmentation, reporting that augmentation reduces the risk of the loss of reduction and screw perforation significantly. In the present study, the complication rate for augmented plating was 8%, comparable to that in Katthagen et al. [14]. Foruria et al. investigated a much larger cohort using the deltopectoral approach. They reported a complication rate of 15% in the cemented group compared to 8.6% in this study. Furthermore, Foruria et al. observed avascular necrosis in 4.8% of the cases [15]. No case of avascular necrosis of the humeral head was observed. This could be due to different surgical approaches. In this study, we only used the minimally invasive delta-split approach. In addition, the surgical time was 34 min shorter in this study compared to that in the studies of Foruria et al., Katthagen et al., and Foruria et al., and this study proved that locked plating with augmentation is a reasonable option with good clinical outcomes and a low complication rate, especially when combined with screw augmentation. Low complication and secondary dislocation rates could even be achieved in AO C-type fractures, which represent 45% of augmented cases. In the literature, these fractures are typically treated with reverse arthroplasty. However, reverse arthroplasty is not without complications. Paras et al. reported tuberosity complications in 25.9% of cases, scapular notching in 18.6%, and heterotopic ossification in 13.2%, and an overall revision rate of 1.8% [16]. Köppe et al. showed that, compared to osteosynthesis, reverse arthroplasty resulted in more hospital adverse events and surgical complications [17].

In their study, Porschke et al. compared the plating of humerus fractures in 31 patients with arthroplasties in 29 patients. They showed that, with the same functional outcome, surgical complications were significantly more frequent after plate osteosynthesis (32.6%) than after arthroplasty (7.2%) [9]. In the present study, a complication rate of 8.6% in augmented locked plating was found and, therefore, comparable to arthroplasty. In addition, the length of stay was, on average, 6 days shorter, and 86.7% of all patients could be discharged either home or to inpatient rehabilitation.

The one-year mortality rate of 9.5% in this study coincides with Lander et al., which, after analysing 42,511 patients following proximal humerus fractures, concluded that surgery is associated with a lower mortality (9.1% vs. 19.9%) [18].

The authors of this study recommend a critical evaluation of all treatment options, especially for the vulnerable geriatric patient group.

In conclusion, minimally invasive modern locked plating with screw tip augmentation is a valuable treatment option for geriatric patients. Compared to the often-cited high complication rates associated with first-generation implants, these complication rates might be comparable to reverse arthroplasty. For this reason, augmented locked plating became the standard treatment of geriatric proximal humerus fractures at our institution.

One limitation of this study is the retrospective design, the postoperative function of which, in contrast to other investigations, was only determined 3 months after surgery and in only 122 of 209 cases, with the understanding that follow up is difficult in the elderly and the expectation that cement has no other influence on the function [9]. On the other hand, in geriatric patients, not only the range of motion but also the loading is important.

Another limitation is that no questionnaires were sent to the patients for follow up to gain additional information, such as that of their quality of life.

The long study period included many different attending surgeons, which may have resulted in some postoperative bias.

The study is also limited by the small number of cases. After screw tip cement augmentation, only 44 patients could be included. Nevertheless, this study is one of the biggest investigations on this issue.

## 5. Conclusions

Screw tip augmentation in osteoporotic proximal humerus fractures leads, in this investigation, to a significantly lower rate of implant-associated complications, especially regarding revision surgery. Therefore, the trend to default to reverse shoulder arthroplasty in geriatric patients with a poor general condition and complex fracture situation should be critically questioned, considering the comparable complication rates in the literature and a less invasive surgery. In these cases, plating might be an alternative to primary reverse shoulder arthroplasty, depending on the fracture morphology and the bone quality.

## Figures and Tables

**Figure 1 jcm-11-06751-f001:**
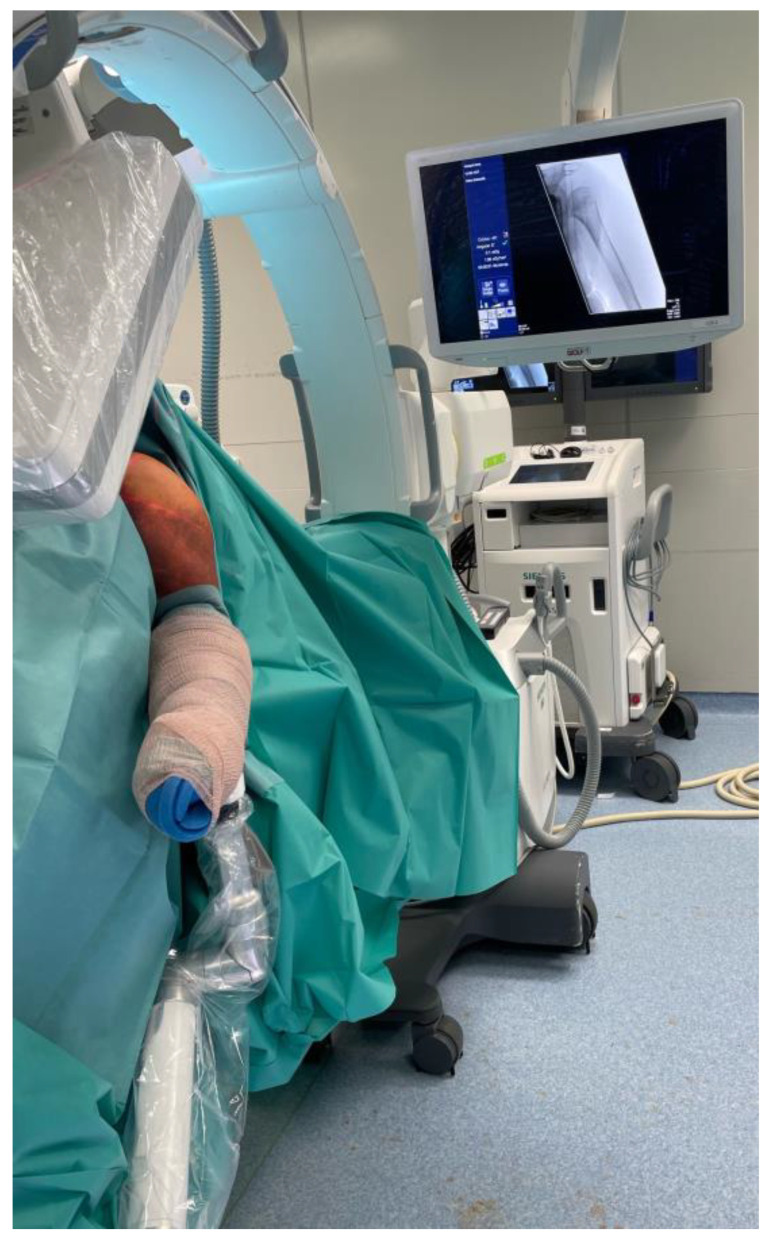
Patient positioning.

**Figure 2 jcm-11-06751-f002:**
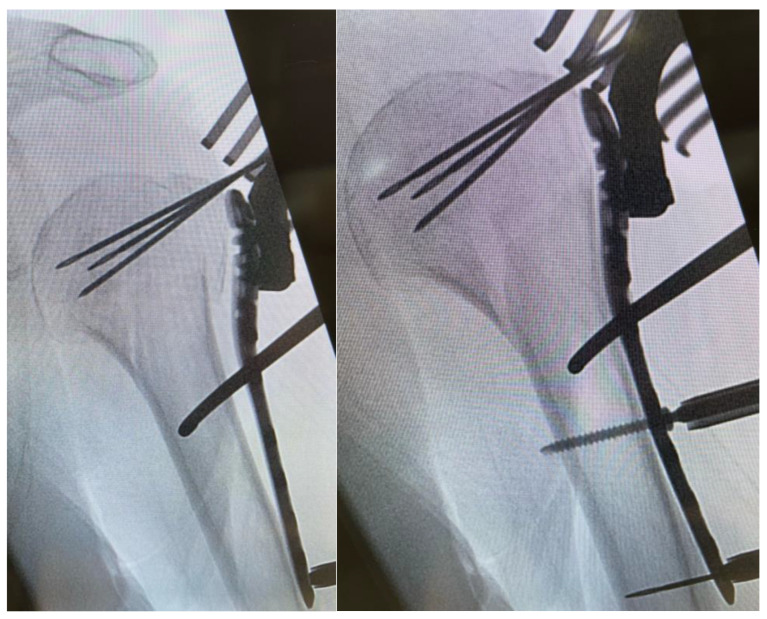
Plate placement, temporary fixation with K-wires and implantation of a cortex screw.

**Figure 3 jcm-11-06751-f003:**
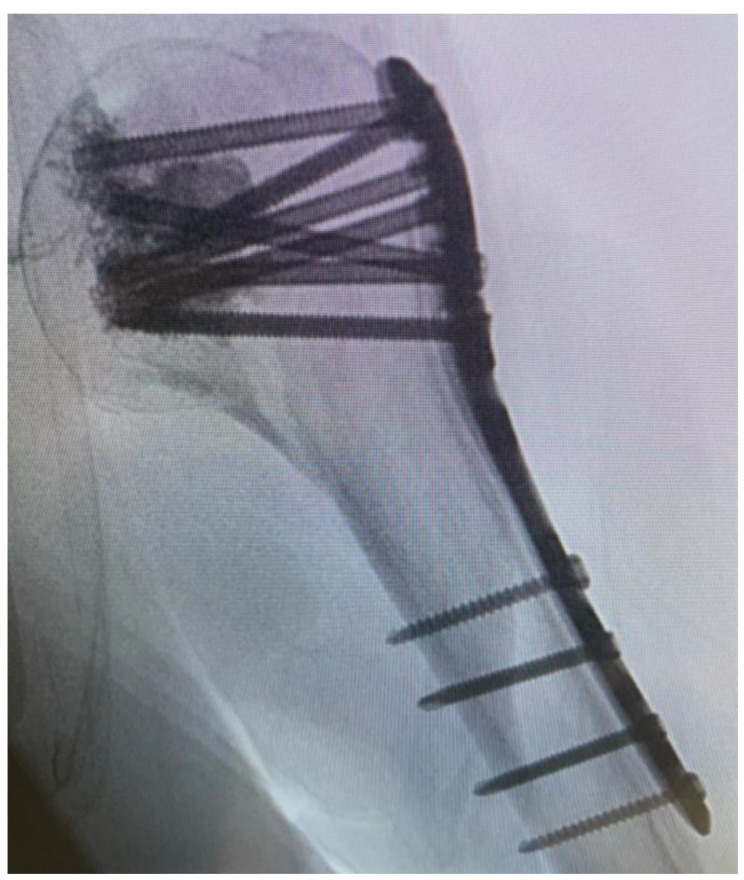
Fluoroscopic control after cement augmentation of the screws.

**Figure 4 jcm-11-06751-f004:**
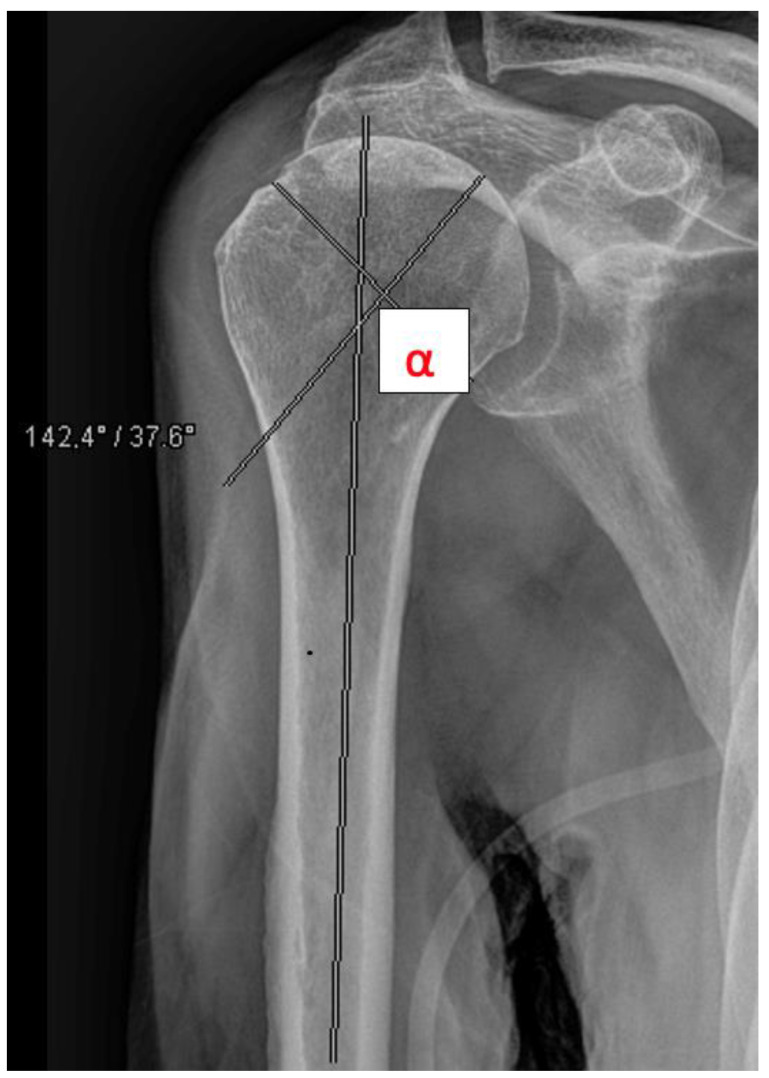
Measurement of the head shaft angle (HAS, e.g., 14.2°).

**Figure 5 jcm-11-06751-f005:**
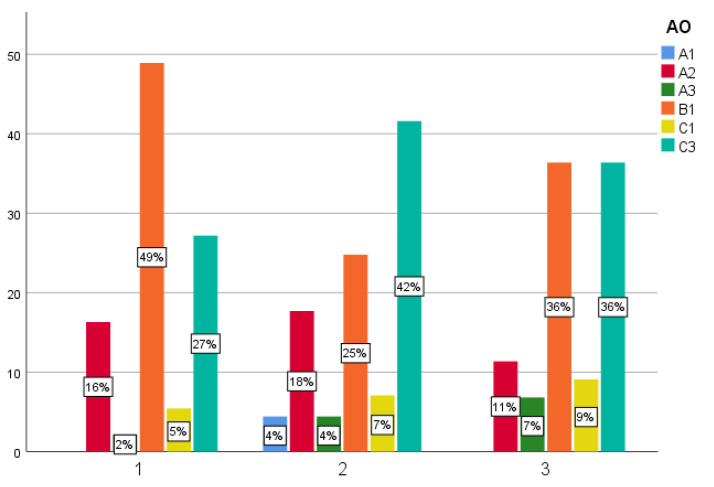
Different treatment options (1 = NCB, 2 = Philos, 3 = Philos-augmented) divided according to the AO classification (*x* axis) in percent (*y* axis).

**Figure 6 jcm-11-06751-f006:**
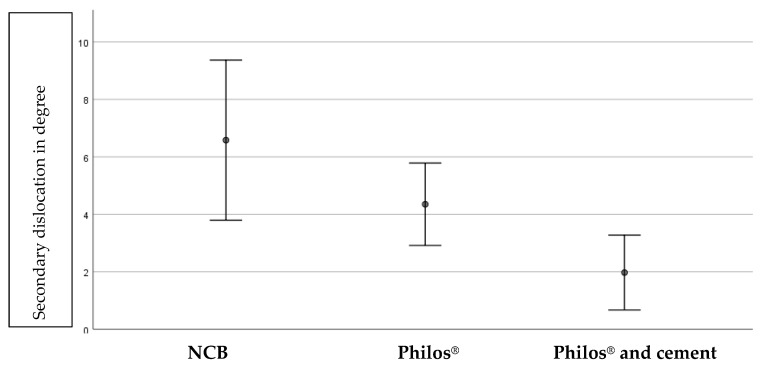
Secondary dislocation in degree (*y* axis) divided into the different type of treatments.

**Table 1 jcm-11-06751-t001:** Inclusion and exclusion criteria.

Inclusion Criteria	Exclusion Criteria
- Proximal humerus fractures in patients over 60 years of age	- Patients under 60 years of age
- Plating with NCB^®^, Philos^®^ and Philos^®^ with cement	- Patients treated with intramedullary nailing or arthroplasty
	- Pathological fracture

## Data Availability

The study did not report any data.

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
