# Peer review of "Feasibility and Radiological Outcome of Minimally Invasive Locked Plating of Proximal Humeral Fractures in Geriatric Patients"

_jcm, 2022, doi:10.3390/jcm11226751_

Round 1
Reviewer 1 Report (Previous Reviewer 2)
I am not sure a retrospective study should change tour practise completley
i am sure there are still instances when repolacement is warrented --bone qulaity etc--i would like to see these exceptions places so readers fdo not think all fall into the fix ORIF category--"most can" but not all and not just based on this study
with this modified alteration to your conclusion
Author Response
Your comment is absolutely correct. Screw tip cement augmentation isn’t the solution for every proximal humerus fracture. There are of course still instances when replacement is necessary, but this study can proof that screw tip augmentation is a valuable method in the treatment of osteoporotic proximal humerus fractures and might be an alternative to arthroplasty in some cases. We tried to underline it in the text.
Reviewer 2 Report (New Reviewer)
Please review spelling and grammar. You say “dislocation” when I think you mean displacement. In your decision to treat these fractures it is unclear why they were all selected for surgery. In addition there are no functional outcomes discussed. . I don’t think simply discussing displacement alone without outcomes is worthwhile. I would recommend adding clarity to your initial selection of treatment pathway and adding functional outcomes data.Author Response
Thank you for your comments! The manuscript was again reviewed by a native English speaker. We also try to underline why patients were chosen for surgery. Also the functional results are discussed in the reviewed version.
The reason why the functional results weren’t discussed till now is that range of motion was only documented in 122 out of 209 patients after 3 months. In the Philos augmented group 50% of the patients achieved at least 90° anteversion and abduction which was only about a third of patients in the other 2 groups (NCB 34.8%, n=46; Philos 35.8%, n=56; augmented Philos 50.0%, n=20). There was no significant difference between the groups.
This manuscript is a resubmission of an earlier submission. The following is a list of the peer review reports and author responses from that submission.
Round 1
Reviewer 1 Report
General –
Although the concept of reporting on newer generation implants for the treatment of proximal humerus fractures is worthwhile there are numerous critical issues in this study. Namely, there are no patient reported outcomes reported for the patients. Radiographic follow-up is limited to 6 weeks which is far too short. It is unclear based off the study what the rates of patients lost to follow-up at 1-year. Even for clinical follow-up 2-years minimum follow-up is typically the standard minimum for “early follow-up” observational studies in orthopedic surgery. The authors would benefit from ensuring the study follows STROBE guidelines for observational studies throughout as there are numerous issues throughout. Overall, the quality of writing could be improved in many areas throughout.
Abstract
Line 25- difference in what?
Line 28- Radiographic follow-up is only available at 6 weeks? This is a very short term frame for this type of study.
Line 30- What is meant by “varus dislocation”
Line 34- Please fix P values to follow AMA formatting throughout
Line 35 – P value missing
Line 39- You can’t make any comparison to rTSA based off this study.
Introduction-
Line 56 – Tuberosities typo
Line 67-69 – Sentence grammatically does not make sense
Please provide a hypothesis for the study
Methods-
Please add surgical description in the methods section
Were these the only exclusion criteria? What about things like revision surgery?
“Retrospective exploratory review” is not a study design
Who performed the radiographic measurements?
Results-
How was “varus collapse” defined.
Line 163 – please provide the results of the regression analysis in more detail.
Why are non-radiographic complications grouped with radiographic complications
Conclusions-
None of the conclusions reported can really be made based off this study. The grouping of radiographic complications (which may or may not be clinically relevant is problematic.
Author Response
Dear ladies and gentleman,
thank you for your extensive comments to improve the quality of our manuscript. The whole text was revised and presented to a native English speaking colleague. In following we try to implement each of your remarks and comment them point by point.
Reviewer 1
General –
Although the concept of reporting on newer generation implants for the treatment of proximal humerus fractures is worthwhile there are numerous critical issues in this study. Namely, there are no patient reported outcomes reported for the patients. Radiographic follow-up is limited to 6 weeks which is far too short. It is unclear based off the study what the rates of patients lost to follow-up at 1-year. Even for clinical follow-up 2-years minimum follow-up is typically the standard minimum for “early follow-up” observational studies in orthopedic surgery. The authors would benefit from ensuring the study follows STROBE guidelines for observational studies throughout as there are numerous issues throughout. Overall, the quality of writing could be improved in many areas throughout.
Thank you for the comments and I absolutely understand your points. In dead (as added under limitations) the postoperative function wasn’t mentioned in this investigation. Main focus was based on implant failures. Knowing that implant loosening in case of proximal humerus fractures is mostly observed 6 weeks after surgery, patients are always seen at this time as out patients. It is really difficult (especially in the geriatric collective) to extend the radiological follow up. This point influenced the design of this retrospective investigation.
Abstract
Line 25- difference in what?
Thank you for the remark. In all groups x rays were performed one day after surgery, showing no difference concerning the head shaft angle. In this case the head shaft angle should be representative for the quality of reduction to achieve a comparability of the 3 groups.
Line 28- Radiographic follow-up is only available at 6 weeks? This is a very short term frame for this type of study.
I absolutely understand your point of view! Having the experience that implant loosening in case of proximal humerus fractures is mostly seen 6 weeks after surgery, patients do always get an appointment 6 weeks after surgery in our out patient department. For one year follow up many patients of the geriatric and immobile collective were contact by phone.
Line 30- What is meant by “varus dislocation”
By using the term “varus dislocation” we mean a decrease of the head shaft angle (normally about 137°) comparing the postoperative and the 6 weeks x rays.
Line 34- Please fix P values to follow AMA formatting throughout
Fix p values were added.
Line 35 – P value missing
Fix p values were added.
Line 39- You can’t make any comparison to rTSA based off this study.
You’re absolutely right! We added that the complication rate of cement augmentated screws in our study is comparable to the in the literature reported complication rate of the RSA.
Introduction-
Line 56 – Tuberosities typo
corrected
Line 67-69 – Sentence grammatically does not make sense
Please provide a hypothesis for the study
I’m with you. We deleted the last sentence and formulated a hypothesis.
Methods-
Please add surgical description in the methods section
You’re right. It doesn’t make sense to mention it under results.
Were these the only exclusion criteria? What about things like revision surgery?
Correct. Revision surgery cases were not included. We added this point.
“Retrospective exploratory review” is not a study design
Right. Corrected.
Who performed the radiographic measurements?
A resident as well as a consultant measured each patient. Added.
Results-
How was “varus collapse” defined.
Varus collapse was defined as varus dislocation leading to surgery.
Line 163 – please provide the results of the regression analysis in more detail.
Why are non-radiographic complications grouped with radiographic complications
Regression analysis is now described in more detail in material & methods. Two different logist regressions were performed for radiographic complications in form of secondary dislocation and for non radiographic complications in form of surgical complications. For all other group comparisons Chi-square test was used.
Conclusions-
None of the conclusions reported can really be made based off this study. The grouping of radiographic complications (which may or may not be clinically relevant is problematic.
I got your point and put my statement into perspective. As already mentioned one limitation is the low number of cases, another is the circumstance that patients with RSA aren’t covered by this investigation. Comparisons are always based on the literature. But our investigation shows that complications, leading to revision surgery, can be reduce significantly by screw tip augmentation. Knowing that many trauma surgeons don’t use this feature, our study can be useful to improve the treatment quality in case of poor bone quality.
Reviewer 2 Report
To really add to the literature this topic and fixation requires a randomized prospective ( maybe even a multicenter ) study
44 patients in the one group are just not enough to make the conclusion you are trying to make at this time
The actual cases are good 'just not powered well enough for a two arm study let alone a 3 arm study
Please keep up this work and develop it so it can add to the literature in a meaningful way
Author Response
Comments and Suggestions for Authors
To really add to the literature this topic and fixation requires a randomized prospective ( maybe even a multicenter ) study
44 patients in the one group are just not enough to make the conclusion you are trying to make at this time
The actual cases are good 'just not powered well enough for a two arm study let alone a 3 arm study
Please keep up this work and develop it so it can add to the literature in a meaningful way
Thank you so much for your comments and your time to review this study.
Of course you are right and it would be a value to have a higher number of cases to improve the meaningfulness. In the literature clinical studies of screw tip augmentation are really hard to find and this study is one of the biggest case series and is able to visualize that complications, leading to revision surgery, can be reduce significantly by screw tip augmentation and shows on the other the development of locking plate systems (NCB vs. Philos). Knowing that many trauma surgeons don’t use the feature of screw tip augmentation, our study can be useful to improve the treatment quality of proximal humerus fractures in case of poor bone quality.
As part of the revision many parts of the manuscript has been improved, particulary the low number of cases has been discussed and was added to the limitations of this study.
Round 2
Reviewer 1 Report
Appropriate changes were made.